# Coexisting ecotypes in long-term evolution emerged from interacting trade-offs

Avik Mukherjee [1,2], Jade Ealy [1,2], Yanqing Huang [1], Nina Catherine Benites[1], Mark Polk [1] & Markus Basan [1] ✉

Evolution of complex communities of coexisting microbes remains poorly understood. The long-term evolution experiment on *Escherichia coli* (LTEE) revealed the spontaneous emergence of stable coexistence of multiple ecotypes, which persisted for more than 14,000 generations of continuous evolution. Here, using a combination of experiments and computer simulations, we show that the emergence and persistence of this phenomenon can be explained by the combination of two interacting trade-offs, rooted in biochemical constraints: First, faster growth is enabled by higher fermentation and obligate acetate excretion. Second, faster growth results in longer lag times when utilizing acetate after glucose is depleted. This combination creates an ecological niche for a slower-growing ecotype, specialized in switching to acetate. These findings demonstrate that trade-offs can give rise to surprisingly complex communities with evolutionarily stable coexistence of multiple variants in even the simplest environments.

The *E. coli* long-term evolution experiment (LTEE) started in 1988 is one of the longest running biological experiments[1,2]. Every morning, about 1% of the population is transferred to a fresh culture medium containing a small amount of glucose, allowing a few generations of growth. The LTEE and similar laboratory evolution experiments appear to select for shorter lag phases when exiting stationary phase, followed by faster growth rates[3]. Yet despite the simplicity of the experimental protocol, the LTEE has resulted in a rich set of surprising phenomena[4]. Among the most unexpected was the spontaneous emergence of coexisting populations of strains, termed ecotypes[5–7]. This effect was originally discovered based on the different colony sizes formed by these strains and they were therefore denoted the L-strain (large colonies) and S-strain (small colonies). Even more surprising than the emergence of this coexistence was its persistence during continued evolution. Although both ecotypes continued to evolve, the coexistence phenotype has been maintained for >14,000-generations, notably without the emergence of a single, tertiary strain that could have invaded and replaced the two coexisting populations[5,6]. Several studies have followed up on these original observations. For example, one study has extensively profiled the ecological and evolutionary dynamics of the L- and S-strain by using reciprocal invasion

experiments with L and S clones from the same or different generation of the LTEE[8].

Theoretical work has also resulted in a better understanding of coexistence phenomena. Evolutionary game theory has shown that coexistence of strains can emerge even in simple environments from negative frequency-dependent selection[9] and analytical solutions for coexistence ratios have been calculated[10]. It has been argued that 'cheating' fermenting and 'cooperating' non-fermenting strains can coexist due to the toxicity of accumulating fermentation products[11]. However, substrate concentration in the LTEE is much too low for this mechanism to play a role and it cannot explain the competitive advantage of the S-strain after glucose is depleted[6]. Neither is there any evidence of specialization of the L-strain and the S-strain for different concentrations of glucose, which has been proposed as a mechanism that can result in coexistence[12]. In this work, we wanted to better understand mechanistically, which physiological constraints result in negative frequency-dependent selection and coexistence on the LTEE.

Rozen and Lenski already performed a phenotypic characterization of the coexisting strains[6] and found that the maximum growth rate in fresh medium of the L-strain was ~20% faster than that of the S-strain in glucose medium. They also showed that the S-strain has a

[1]Harvard Medical School, Department of Systems biology, 200 Longwood Avenue, Boston, MA 02115, USA. [2]These authors contributed equally: Avik Mukherjee, Jade Ealy. ✉e-mail: markus@hms.harvard.edu

fitness advantage in glucose-depleted medium. By using medium conditioned by the L-strain, they showed that this fitness advantage is based on nutrients left behind by the L-strain. However, it remains unclear to this day why coexistence emerged in the first place and why it proved remarkably persistent under continued evolution. This lack of understanding is partly due to the difficulty of studying constraints of long-term evolutionary processes experimentally.

In this work, to overcome this challenge, we employ a combination of experiments and computer simulations. We hypothesized that coexistence results from a combination of two recently uncovered trade-offs. The first is a trade-off between growth rate and fermentation[13,14] (see Fig. 1a). It was recently demonstrated that fermentation results in acetate excretion and enables faster growth because it requires a smaller amount of protein investment per flux of ATP produced[13,14]. Thereby, the 'leaner' fermentation pathway frees up proteomic resources that enable faster replication, including a higher ribosomal proteome fraction. Indeed, forcing cells to ferment directly

results in faster growth rates[14]. Mathematically, this trade-off between the growth rate $\lambda$ and the acetate excretion rate $j_{ace}$ is determined by a Pareto front, described by the following expression[13]:

$$j_{ace} \geq \alpha(\lambda - \lambda_0)\theta(\lambda - \lambda_0), \qquad (1)$$

where $\theta$ is the heaviside function, and the proportionality constant $\alpha$ and the threshold growth rate $\lambda_0$ have been previously empirically determined from the strain NCM3722[13].

The second independent trade-off required for coexistence is a trade-off between growth rate and ability to quickly switch to growth on acetate (see Fig. 1b). Faster growth results in longer lag phases when shifting from glycolytic to gluconeogenic substrates such as acetate[15,16]. These lag times are remarkably long because of the difficulty of reversing the direction of glycolytic flux due to key irreversible reactions in glycolysis[15,16]. This second trade-off is also characterized by

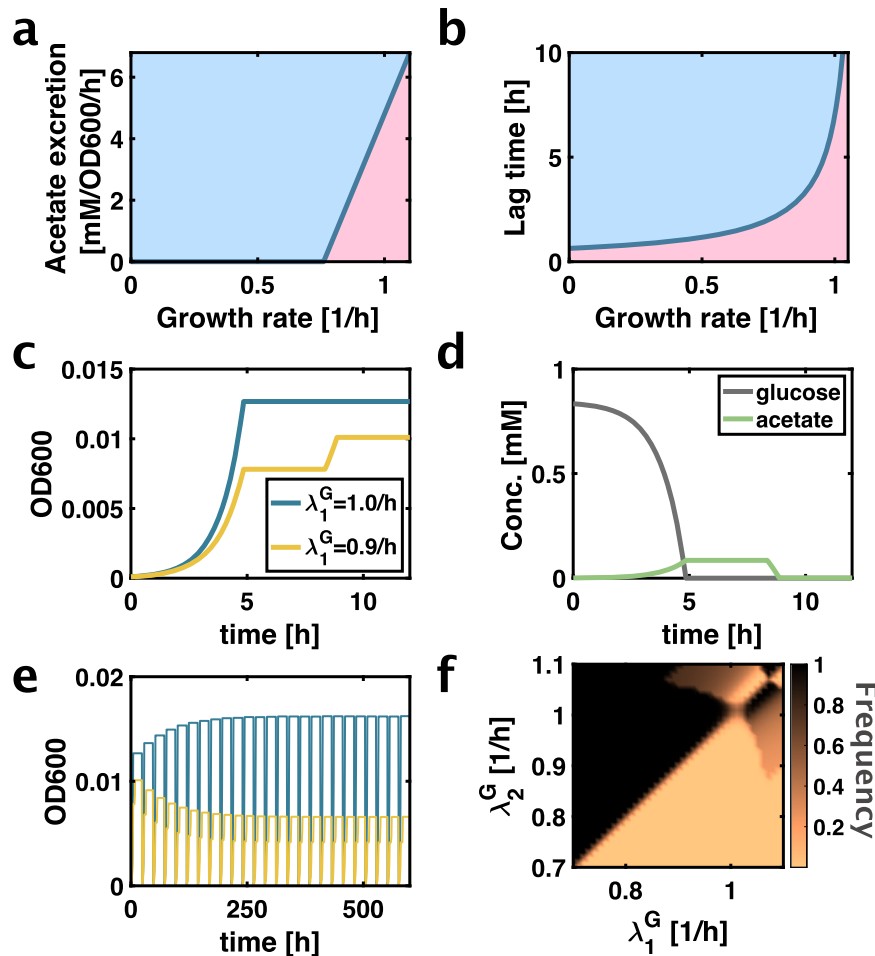

**Fig. 1 | Combination of two trade-offs results in coexistence of strains on single substrate. a** Trade-off between growth rate and acetate excretion rate (in units of carbon atoms). Solid line is the Pareto front between the accessible phenotypes (blue) and inaccessible phenotypes (red). Parameters were previously measured for the NCM3722 strain[13]. **b** Trade-off between growth rate and lag time to acetate. Solid line is the Pareto front between the accessible phenotypes (blue) and inaccessible phenotypes (red). Parameters were previously measured for NCM3722 strain[15]. **c** Growth dynamics of two strains after dilution. The slower growing strain can resume growth on acetate more quickly. **d** Glucose and acetate concentrations in the simulation corresponding to panel c. Concentrations are in units of carbon atoms. Acetate accumulated during growth on glucose. After glucose is depleted, neither strain can grow due to their respective lag phases. The slower growing strain exits lag phase more quickly and consumes acetate before

the faster growing strain can exit lag phase. **e** Abundances of the two strains in many consecutive daily dilutions. After a few days, the abundances of the two strains do not change anymore with additional dilutions, indicating stable coexistence. **f** Steady-state frequency (colour bar) reached by the daily dilution simulation with two strains of different glucose growth rates: $\lambda_1^G$ and $\lambda_2^G$. Frequency of strain 2 is plotted, hence a frequency of 0.8 means that 80% of the population is strain 2 and 20% is strain 1. A frequency of 0 or 1 means no coexistence, as is found for slow growth rates of either strain (lower / left side of the plot). We assume that the phenotypes of the strains are directly on the Pareto fronts given by Eqs. [S2, S7]. Therefore, strain phenotypes are uniquely determined by their glucose growth rates and as a result the plot is symmetric. Source data is provided as a source data file.

a Pareto front, which relates the lag time $T_{lag}$ with the growth rate $\lambda$:

$$T_{lag} \geq \beta / (\lambda_C - \lambda), \tag{2}$$

where the coefficient $\beta$ and the critical growth rate $\lambda_C$ have also been also determined empirically from the strain NCM3722[15]. The critical growth rate $\lambda_C$ is the growth rate at which lag times diverge and switching becomes impossible. We note that the lag time trade-off given by Eq. [2] is independent and mechanistically distinct from the acetate excretion trade-off given by Eq. [1]. The acetate excretion trade-off results from different protein cost of metabolic pathways for ATP production[13]. The lag time trade-off results from the difficulty of reversing flux in glycolysis while preventing futile cycling due to effectively irreversible metabolic reaction in glycolysis[15]. The phenotypic combinations allowed by the trade-offs given by Eqs. [1–2] can be visualized in a 3D plot of growth rate, acetate excretion rate and lag time (see Fig. S1). The intersection of the two Pareto surfaces is a line, plotted in Fig. S1.

We hypothesized that the combination of these two independent trade-offs can result in negative frequency-dependent selection leading to stable coexistence of two strains. The faster growing strain produces more acetate according to Eq. [1]. Therefore, an excess of the faster growing strain results in more acetate in the medium after growth on glucose, which favors growth of the slower growing strain that is more quickly able to utilize acetate, according to Eq. [2]. Conversely, if the abundance of the fast-growing strain is low, according to Eq. [1], there is less acetate in the medium for the slow-growing strain to utilize after glucose is depleted, reducing the benefit of the slow growing strain from quickly switching to acetate. But growth on glucose favors the faster growing strain. Hence, this situation results in stable coexistence of the two strains.

## Results

### Simulation of coexistence

To test if this hypothesis was theoretically correct, we implemented a simple computer simulation of the LTEE (see Supplementary Note 1 for mathematical details). Just as in the LTEE, every 24 h, a 100-fold dilution in glucose medium was introduced in the simulation. In the simulation, the two strains grow at their respective growth rates (Fig. 1c) and consume glucose according to their respective biomass yield (Eq. S5), which is determined by their respective acetate excretion rates. Acetate accumulates in the medium as glucose is consumed (Fig. 1d). When glucose is depleted, cells undergo a lag phase without growth, whose duration is restricted by Eq. [2], before they resume growth on acetate (Fig. 1c).

When we simulated these dynamics, we indeed found coexistence for certain parameter combinations (Fig. 1e). If the phenotypes of both strains (i.e., growth rates, acetate excretion rates and lag times) are directly on the two Pareto fronts to maximize their fitness, these phenotypes are fully characterized by their respective growth rates on glucose. We wanted to determine for which combinations of growth rates there exists stable coexistence. Therefore, we performed a parameter scan and simulated the serial dilution experiment with different combinations of strains with various growth rates. We let the simulation run until the populations reached stable steady-state levels. We plotted the resulting coexistence frequency (colour code) as a function of the growth rates of the two strains (two axes) (Fig. 1f). The resulting plot is symmetric because the two strains are interchangeable, as each strain is uniquely determined by its growth rate. A frequency of either 0 or 1 indicates no coexistence, as the population consists only of one strain. When both strains grow at slow growth rates below the acetate excretion threshold given by $\lambda_0$, there is no acetate excretion according to Eq. [1]. In this regime there is no coexistence (as indicated by frequencies of either 0 or 1). However, for faster growth rates, we indeed found that there is a large parameter

regime where coexistence generally emerges. Interestingly, because evolution in the LTEE prior to the emergence of coexisting ecotypes favors faster growth rates, we expect the LTEE to generally move into the fast growth rate parameter regime that leads to coexistence of two strains. Hence, rather than constituting a special situation, coexistence of two strains could be the general outcome of the LTEE, based on the trade-offs given by Eqs. [1–2].

### In silico evolutionary dynamics

Next, we wanted to test in silico evolutionary dynamics resulting from this theoretical model. We therefore introduced a third strain in the system. While two strains coexisted for many parameter combinations, the introduction of a distinct third strain always resulted in elimination of one of the strains. To implement evolution, whenever the abundance of one of the strains dropped below a low threshold, indicating elimination from the population, a new strain whose growth rate $\lambda$ was randomly chosen from the interval $[0, \lambda_C]$ was introduced. To speed up the simulation, we again assumed that phenotypes were directly on the two Pareto fronts because phenotypes away from the Pareto fronts are sub-optimal. Running this evolutionary simulation resulted in subsequent rounds of invading strains (Fig. 2a) and evolving strain phenotypes. The simulation converged to the same specific combination of strains (Fig. 2b, Fig. S2). When we initiated the simulation with this 'stable' combination of strain growth rates and simulated additional 50,000 days of evolution, where randomly sampled strains could invade the population, substantially different strains were unable to replace either original strain in the population, and the dynamics were soon dominated by strains that were only incrementally different from the stable strains. This indicates that this combination of growth rates is an evolutionarily stable configuration (Fig. 2c). We next ran the evolutionary simulation 10,000 times for 50,000 simulated days, each time initialized with a random combination of three strains growing at different growth rates. These randomly chosen initial growth rates cover the space of possible initial configurations (Fig. 2d). Independent of the initial conditions, the evolutionary simulation converged to the same combination of two growth rates, as seen in the clusters of the final growth rates of the simulation in Fig. 2e and also in the 2D projections of this plot shown in Fig. S2b–d. These data suggest that this combination of strains is the only stable-fixed point of the evolutionary dynamics that we simulated.

### Phenotypic characterization

Next, we wanted to know to what extent the model predictions were reflected in the real phenotypes of the L- and S-ecotypes that emerged from the LTEE. We therefore phenotypically characterized a set of L- and S-ecotypes isolated at a specific timepoint in the LTEE. First, we measured acetate excretion rates of the L- and S-strains by taking supernatant samples along the growth curves of the L- and S-strain. Indeed, we found that different growth rates of the L- and S-strains on glucose also resulted in different acetate excretion rates (Fig. 3a), consistent with the trade-off between growth rate and acetate excretion rate (Fig. 1a).

To confirm the role of the second trade-off, we next wanted to measure lag phases of the L- and S-strains. The amount of acetate excreted with the glucose concentrations used in the LTEE is very low, making it difficult to quantify lag phases. Therefore, we grew the two strains separately in the presence of the normal LTEE concentration of glucose but with a much greater concentration of acetate present in the medium to measure their respective lag times in the resulting diauxic shift. To get a more precise readout of growth at the low optical densities, where glucose is depleted, we also transfected the L- and S-strains with fluorescent proteins. Indeed, precisely as expected from the model, the faster growth rate of the L-strain on glucose resulted in a longer lag time in the diauxic shift, as compared to the S-strain (Fig. 3b). This explains how the S-strain derives its fitness

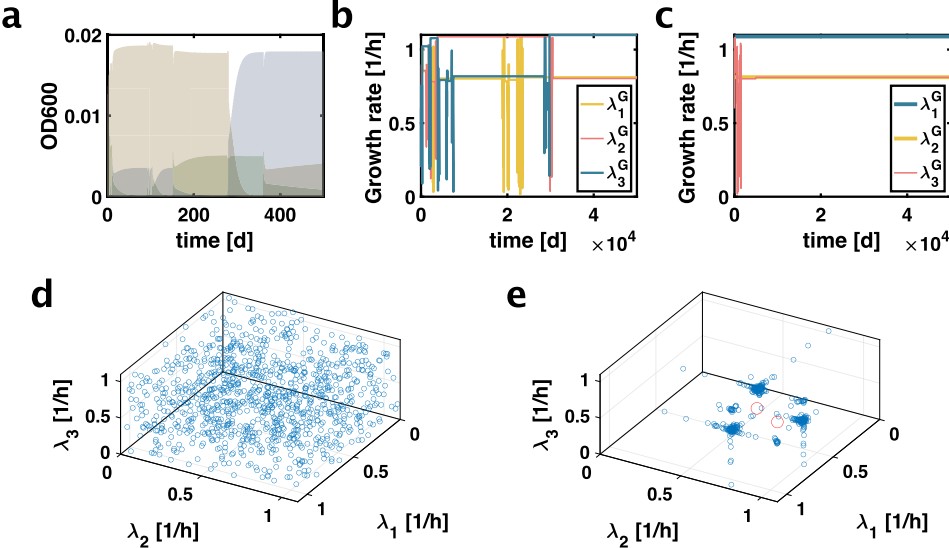

**Fig. 2 | Convergent evolutionary dynamics in coexisting populations. a** OD600 versus time for three strains (different colours). While two strains coexist for many combinations of growth rates, with three strains one strain is always eliminated. When the abundance of one of the three strains drops below a low threshold indicating elimination, a new strain is introduced with a randomly selected growth rate. These dynamics result in subsequent rounds of invasion and elimination. **b** Growth rates of three strains in the long-term evolutionary dynamics simulation introduced in panel a. Strain growth rates uniquely determine all strain phenotypes in our model because we assume phenotypes are directly on the Pareto fronts given by Eqs. [S2, S7]. The evolutionary dynamics converges to one combination of strains wherein one strain approaches the maximum critical growth rate $\lambda_C$. In this combination of strains, the second strain grows more slowly than the first, yet somewhat faster than the acetate excretion threshold growth rate $\lambda_0$. An additional repeat of the simulation that converged to the same combination of growth rates is shown in Fig. S2a. **c** Initiating the simulation with these steady-state growth rates, new strains were only able to invade if their growth rates were extremely close to the initial growth rates, indicating proximity to a stable fixed point of the dynamics.

**d** To test if this combination of two growth rates constituted the only fixed point of the evolutionary dynamics, we ran the evolutionary simulation 10,000 times. Each simulation was run with a randomly selected combination of initial growth rates. Each combination of initial growth rates for each of the 10,000 simulations is represented by one blue point in the 3D plot shown in **d**. **e** Each blue data point represents one combination of final growth rates of one simulation after 50,000 simulated days of evolution. The final growth rates cluster at combinations of the same two growth rates found in **b** and **c**. This also becomes apparent in the 2D projections of this plot shown in Fig. S2b, c. We note the third growth rate also tends to be close to one of the two fixed point growth rates because this results in slow elimination dynamics of the less fit strain and the simulation is more likely to terminate in such a configuration. The red spheres indicate a lack of clusters for three identical growth rates, which cannot be seen from the 2D projections in Fig. S2. Together, these data indicate this combination of two strains at these specific growth rates (see **b**, **c**), constitutes the sole stable fixed-point of the evolutionary dynamics simulation. Source data is provided as a source data file.

advantage after glucose depletion in the LTEE, whereas the L-strain benefits from faster growth enabled by higher acetate excretion during the glucose growth phase. This interplay is summarized in Fig. 3c.

## Coexistence and break-down of coexistence in daily dilution experiment

Finally, we utilized the L- and S-strains with two different fluorescent proteins to perform the daily dilution experiment (see Fig. S3 for detailed protocol). We initiated the experiment with different ratios of the L- and S-strain and every 24 h, performed a 100-fold dilution in minimal medium with the same glucose concentration as the LTEE. As expected, the different cultures reached stable coexistence after several days (Fig. 4a). We next wanted to directly test the causal requirement of the two trade-offs for coexistence. Therefore, we split coexisting cultures and continued daily dilution in different modified media. In one case, a large amount of acetate (5 mM) was provided in addition to the normal glucose amount (139μM) with every dilution. According to the simulation, for a high enough amount of acetate provided in the medium, coexistence should disappear, and the L-strain should be eliminated. This is because when acetate is provided in the medium, the S-strain does not need to rely on the presence of the L-strain with its higher acetate excretion rate (Fig. S4). Indeed, experimentally, serial dilution in medium with high concentrations of acetate resulted in elimination of the L-strain (Fig. 4b, c squares). Conversely, at slower growth rates without acetate excretion, the simulation predicted absence of coexistence (see Fig. 1f). Indeed, performing the daily dilution experiment in glycerol minimal

medium, where no acetate is excreted[13], coexistence disappeared, and the slower-growing L-strain was eliminated (Fig. 4b, c triangles and Fig. S5).

## Discussion

Our findings show that the emergence and persistence of coexistence in the LTEE can be explained by the interplay of two independent trade-offs given by Eqs. [1–2]. Despite thousands of generations of evolution, we show that the phenotypes of the L- and S-strain remain constrained by the two trade-offs (see Fig. 3a–c): The L-strain is a fermentation specialist that grows more quickly on glucose by excreting more acetate. Conversely, the S-strain is specialized in switching to acetate at the cost of a slower growth rate on glucose.

The long-term persistence of coexistence in the LTEE under continued evolution is remarkable because a hypothetical fast-growing strain that does not excrete acetate, having overcome the acetate excretion trade-off by evolution, would take over the population and abolish coexistence. Similarly, a hypothetical fast-growing strain without a lag when switching to acetate, having overcome the lag time trade-off by evolution, would immediately be able to utilize excreted acetate as efficiently as the slower growing strain. Therefore, this hypothetical strain would also outcompete both the L- and S-strain and abolish coexistence. Both trade-off are required for coexistence to be maintained. However, neither of these hypothetical scenarios have occurred in the LTEE. This demonstrates that the two trade-offs are indeed evolutionary constraints that are not easily overcome even after thousands of generations of evolution.

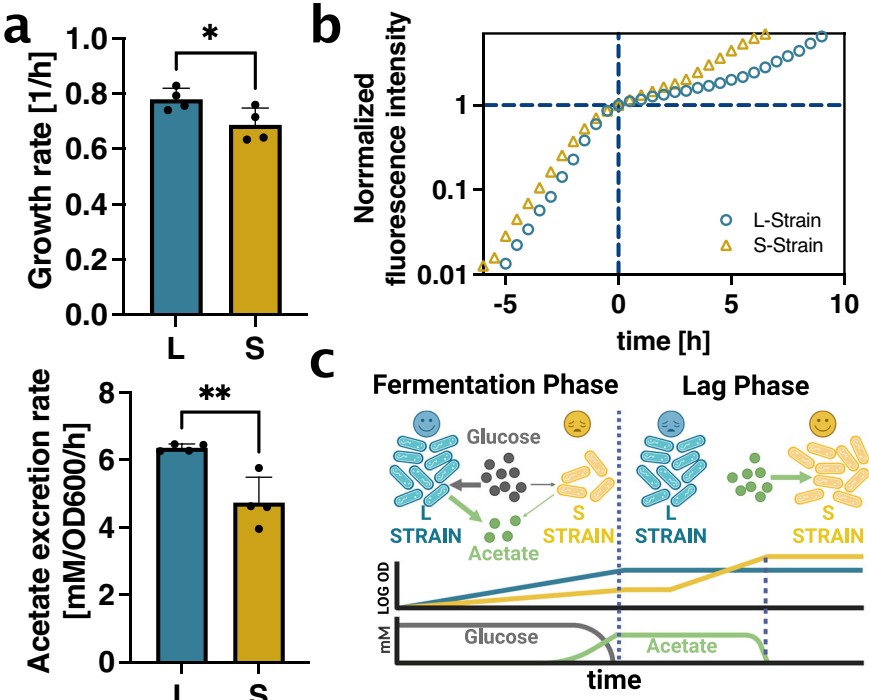

**Fig. 3 | Phenotypic characterization of L- and S-strain. a** Single-strain growth rates in glucose minimal medium batch culture (upper panel, $n = 4$ biological replicates, unpaired $t$-test was performed, $P$-value = 0.044) and corresponding acetate excretion rates (lower panel, $n = 4$ biological replicates, unpaired $t$-test was performed, $P$-value = 0.0052). Error bars represent standard deviation. **b** Diauxic shift from glucose to acetate. Medium contains typical glucose concentration of LTEE (139μM) with additional acetate (10 mM). $n = 2$ replicate (technical) well for each strain used and mean values plotted, for details please refer to method. **c** Schematic illustration of the interplay of acetate excretion and lag times resulting in coexistence. The L-strain is a fermentation specialist, enabling faster growth on glucose via acetate excretion (left). The S-strain is specialized in switching to acetate due to its slower growth rate on glucose (right). Source data is provided as a source data file.

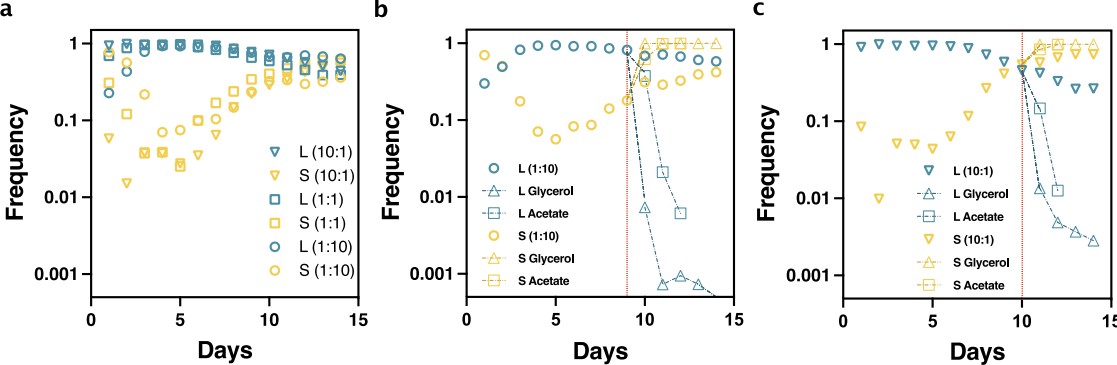

**Fig. 4 | Coexistence and breakdown of coexistence of L and S ecotypes. a** Stable coexistence of L and S ecotypes occurred irrespective of initial inoculation ratios of these ecotypes (L:S = 10:1/1:1/1:10). For each curve, frequency is averaged from two independent experiments; for details, please refer to methods. **b** After establishing coexistence, the culture (from L:S = 1:10 tube of first replicate) was split on day 9 and daily dilution was continued in new media in duplicate tubes. In one case, a high amount of acetate (5 mM, squares) was added in addition to the normal glucose medium (139 mM) at every dilution. In the other case, glucose was replaced by an equivalent amount of glycerol (triangles) as the carbon source. In both cases, coexistence quickly disappeared. As a control, the original experiment was continued in parallel in regular daily dilution medium (circles). For new media conditions, average frequencies from duplicate tubes are plotted, for details, please refer methods. **c** In another experiment, when the L:S = 10:1 culture from second replicate reached stable coexistence, it was also split in acetate (squares) and glycerol (triangles) media in duplicate, and the data was plotted in the same way as in panel b. As in both **b** and **c**, the L-strain was lost quickly in acetate (squares) and in glycerol (triangle), whereas coexistence persisted in the parental control tube (inverted triangles). Source data is provided as a source data file.

## Methods
### Bacterial strains
Strains used in this paper are the original *E. coli* strains isolated by Rozen and Lenski[6]. Original S-strain used in this study is REL7409 (which is a S morph clone from REL7324). Original L-strain used in this study is REL 7410 (which is a L morph clone from REL7324). Strains were received as a kind gift from Dr. Richard Lenski.

### Preparation of L strain expressing cytoplasmic CFP (Cyan fluorescence protein) and S-strain expressing cytoplasmic YFP (Yellow fluorescence protein)
RNA1 promoter driving CFP:: Kanamycin$^r$ was transferred to L-strain via P1 transduction. RNA1 promoter driving mVenus:: Kanamycin$^r$ was transferred to S-strain via the same protocol and L-strain expressing cytoplasmic YFP was also prepared

by same protocol. P1 transduction protocol was adapted from Thomason et al.[17].

## Growth media

N+C+ minimal medium with 20 mM $NH_4Cl$ and different concentrations of carbon sources was used. Composition of N+C+ minimal medium is detailed in table below. Each medium was prepared using a 4X N-C- salt solution stock, which was diluted to 1X concentration in final media formulation and supplemented with 20 mM $NH_4Cl$ (final concentration) and the carbon source. After preparation, each medium was filtered using disposable vacuum filtration system (Corning, PES filter with 0.22 pore size).

1 L of 4x N-C- stock salt solution was prepared in the following way:

| Species | Grams (g) | Formula weight (Da) | Molarity (mM) |
|---|---|---|---|
| $K_2SO_4$ | 4 | 174.26 | 23.0 |
| $K_2HPO_4$ | 54 | 174.18 | 310.0 |
| $KH_2PO_4$ | 18.8 | 136.09 | 138.1 |
| $MgSO_4$ | 0.192 | 120.37 | 1.6 |
| NaCl | 10 | 58.44 | 171.1 |

$K_2SO_4$ (VWR BDH Chemicals, Catalog #: BDH4618-500G), $K_2HPO_4$ (VWR BDH Chemicals, Catalog #: BDH9266-2.5KG), $KH_2PO_4$ (VWR BDH Chemicals, BDH9268-2.5KG), MgSO4 (Sigma Aldrich, M7506-1KG), NaCl (Sigma Aldrich, S7653-1KG)

Carbon sources: D(+) Mannose (Sigma Aldrich, M6020-25G, Lot #BCBV4824), Glucose (Sigma, G5146), Sodium Acetate (Sigma S2889), Glycerol (VWR BDHH1172-1LP)

## Measurement of Acetate excretion rate

For acetate excretion rate measurement, the original S-strain (REL7409) and original L-strain (REL 7410) were used and excretion rate was measured following the protocol previously described by Basan et al.[13]. Briefly, L- and S-strain were inoculated in LB medium from single colonies and then diluted in N+C+ minimal medium with 44.4 mM glucose and 20 mM $NH_4Cl$ for overnight culture. The next morning, tubes with fresh N+C+ minimal medium with 44.4 mM glucose and 20 mM $NH_4Cl$ were inoculated from saturated overnight culture. Freshly inoculated tubes were placed in shaking water bath incubator at 37 °C, with 200 RPM orbital shaking. Optical densities (OD600) of the freshly inoculated cultures were measured using a spectrophotometer (Genesys 30 visible spectrophotometer, Thermo Scientific). After the freshly inoculated cultures reached exponential growth phase, 1 ml culture samples were pipetted out from the inoculated tubes in 1.5 ml microcentrifuge tubes. Multiple samples were taken from each tube along exponential growth phase and optical density at the time of sample collection was noted. After sample collection, bacterial cells were precipitated out of the aspirated samples by centrifuging the 1.5 ml tubes in a microcentrifuge at 16000Xg for 3 min. Media supernatants from the tubes were carefully transferred to fresh 1.5 ml tubes, without disturbing the pellet. Acetate concentrations in the collected media samples were measured using an acetate colorimetric assay kit (Sigma-Aldrich, MAK086) following the manufacturer's protocol. Absorbance at 450 nm was read in a BioTek, Synergy H1 microplate reader. Measured acetate concentrations were plotted as a function of optical density of the samples at the time of sample collection. Acetate excretion rate was calculated from the slope of the linear regression through the origin, multiplied by the growth rate as described before by Basan et al.[13].

## Diauxic shift measurement

L-strain expressing cytoplasmic YFP and S-strain expressing cytoplasmic YFP were grown in LB culture from single isolated colonies. For overnight culture, N+C+ minimal media with 20 mM $NH_4Cl$ and 139 µM D(+) glucose (concentration used by Lenski in the LTEE) was used. Strains were separately inoculated from LB culture in 5 ml media and inoculated tubes were placed in a shaking air incubator (Infors HT) maintained at 37 °C, with 200 RPM orbital shaking. The next day, 1% of the saturated overnight cultures were freshly inoculated separately in 1 ml fresh N+C+ minimal media with 20 mM $NH_4Cl$, 139 µM D(+) glucose and 10 mM sodium acetate and mixed well by vortex mixing. From the freshly inoculated tubes, 200 µl samples were transferred to a clear bottom 96 well plate with dark wall (Greiner bio-one, 655090). Two separate wells of the 96 well plate was used for each strain. Fluorescence intensity of YFP from each well was measured in 30 min intervals for 24 h in a BioTek Synergy H1 microplate reader, maintained at 37 °C with continuous shaking. Fluorescence intensity from duplicate wells was calculated and normalized to the point when the exponential growth slowed down for each strain, indicating the depletion of glucose and initiation of diauxic shift. Time of normalization for each strain was denoted as time 0 and normalized average fluorescence intensity calculated from duplicate wells as a function of time was plotted in semi-log scale.

## Coexistence experiment

For coexistence experiment, we used L-strain with cytoplasmic CFP (henceforth mentioned as LC) and S-strain with cytoplasmic YFP (henceforth referred to as SY). LC and SY strains were grown in LB for 3 h, then diluted in 5 ml N+C+ minimal medium with 20 mM $NH_4Cl$ and 139µM D(+) glucose (henceforth referred to as daily dilution medium) from isolated single colonies, and grown overnight in a shaker air incubator (Infors HT, maintained at 37 °C, with 200 RPM orbital shaking). For two replicate experiments, two separate colonies of LC and SY strain were inoculated in LB and subsequently transferred to overnight culture in minimal medium. On the next day, we inoculated 5 ml tubes with fresh daily dilution medium with the different overnight cultures. We mixed LC and SY strains at different initial ratios and ensured that the total inoculum volume was 1% of the medium (i.e. 50 µL in 5 ml medium). We started the experiment with three initial ratios of LC and SY strains (LC:SY = 10:1, 45.5 µL LC and 4.5 µL SY; LC:SY = 1:1, 25 µL LC and 25 µL SY; LC:SY = 1:10, 4.5 µL LC and 45.5 µL SY). After inoculation, tubes were transferred to an air incubator (Infors HT, maintained at 37 °C, with 200 RPM orbital shaking). Every 24 h, we transferred 1% from the saturated overnight culture to new tubes containing daily dilution medium for daily dilution. Samples for colony counting were taken from the overnight saturated co-culture tubes and diluted 10,000-fold in N+C+ minimal medium. 100µL of diluted culture was plated on LB-Agar plates (in triplicate) for colony counting purpose. After plating, LB-Agar plates were incubated overnight in a 37 °C incubator. The next day each plate was imaged using a custom-built fluorescence colony imager set up. Each plate was imaged in three channels (brightfield, CFP-fluorescence channel and YFP-fluorescence channel). The number of LC (CFP colonies, representing L-strain) and SY (YFP colonies, representing S-strain) colonies from each plate was determined using Stardist plugin in Fiji (imageJ) using a custom macro. After automatic count, we also manually inspected each image for obvious errors, such as false positive or false negative colony detection. If required, total counts were rectified. Frequency of LC and SY colonies were plotted as a function of time.

As mentioned, plating was done in triplicate from each tube for each biological replicate experiment. For each ratio conditions of individual biological replicate, frequencies of L- and S-strain were calculated from each plate of the triplicate plating and then averaged. After this we averaged the frequencies from two biological replicate

experiments. Finally, we plotted the mean frequency for each ratio from two biological replicates.

## Coexistence breaking experiment

After the LC:SY = 1:10 batch culture from first replicate of the initial coexistence experiment was plated for consecutive 9 days and showed stable coexistence, we split this culture into different media conditions. We continued diluting the original batch culture in daily dilution medium each day, while transferring 1% inoculum from the coexisting stable batch culture to the following media in duplicate tubes:

1. N + C+minimal media with 20 mM $NH_4Cl$ and 139μM D(+) Glucose and 5 mM Sodium Acetate (henceforth referred to as Acetate-media),
2. N + C+minimal media with 20 mM $NH_4Cl$ and 277.52 μM Glycerol (we kept the number of carbon atom the same as in daily dilution medium. Glycerol being a 3 C compound was added in double molar amount to that of glucose).

Each day 1% inoculum was transferred to the respective type of fresh medium. Colony counts of L- and S-strains on each day in these new media conditions were determined by plating from each tube in triplicate plates. First, for each tube with new media condition, we calculated the frequency of L- and S-strain from each plate of the triplicate plates and averaged. Then frequency calculated for each of the duplicate tubes were averaged. We observed that the optical density of the culture reached a six-fold higher level in acetate as compared to the daily dilution medium. Thus, for colony counting, we first diluted the overnight acetate culture 6-fold, and then performed a 10,000-fold dilution and plated 100 μL sample (in triplicate) for colony count. (Another replicate of the coexistence breaking experiment was performed when the LC:SY = 10:1 from the second biological replicate reached steady-state coexistence. That culture was also split following the same protocol described above).

## Statistical methods

Statistical tests were performed using GraphPad Prism. For Fig. 3a and Fig. S5, unpaired t-tests were performed and obtained *p*-values are reported in the respective figure legends.

*P*-value < 0.0001 represented as: ****, *P*-value ranging from 0.0001 to 0.001 denoted as: ***, *P*-value ranging from 0.001 to 0.01 denoted as: **, *P*-value ranging from 0.01 to 0.05 denoted as: *, and *P*-value ≥ 0.05 denoted as not-significant or ns wherever necessary.

## Schematic graphics

Schematic graphics used in some figures (Fig. 3 and Fig. S3) is prepared with biorender.com.

## Reporting summary

Further information on research design is available in the Nature Portfolio Reporting Summary linked to this article.

## Data availability

Source data are provided as a Source Data.ZIP file. Source data are provided with this paper.

## Code availability

The simulation code will be shared upon request.

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

## Acknowledgements

We would like to thank Dr. Richard Lenski for kindly sharing the LTEE ecotypes with us and Dr. Oskar Hallatschek helpful discussions and for bringing the coexistence phenomenon in the LTEE to our attention. We would also like to thank Daniel Eaton and Johan Paulsson lab for providing fluorescent protein expressing strains used as donor strain for P1 transduction. This project was supported by MIRA grant (5R35GM137895), the Systems, Synthetic and Quantitative Biology PhD training award (T32GM135014), and an HMS Junior Faculty Armenise grant to M.B.

## Author contributions

A.M., M.B. conceptualized the project and designed the experiments. A.M., M.B. performed the modelling and simulation. J.E., A.M., Y.H., performed the coexistence experiment and analyzed data. A.M., Y.H. performed lag phase measurement and other experiments and analyzed the data. Y.H. prepared the Image processing pipeline. N.C.B. and M.P. contributed to dilution experiments and analyzed the data. M.B., A.M., J.E., N.C.B., Y.H., M.P. contributed to preparing the manuscript.

## Competing interests

The authors declare no competing interest.
