## [Peer Review File · Nature Communications]

Coexisting ecotypes in long-term evolution emerged from interacting trade-offsREVIEWER COMMENTS

Reviewer #1 (Remarks to the Author):

In this contribution, the authors use a computational simulation of bacterial growth to demonstrate that the coexistence of two evolved strains of *E. coli* bacteria (obtained from the LTEE) can coexist due to competing tradeoffs. I find the study to be well done, the arguments are convincing, and the computational simulation appears to be working well.

There are a number of things that I would like the authors to clarify. And I also have a suggestion for a type of analysis that might clarify some aspects of the dynamics.

First, I was wondering why the authors performed the evolutionary simulation only in triplicate. I would imagine that with today's computers you could get much larger numbers, so that it is possible to quantitatively test a hypothesis, as opposed to writing: "it converged each time".

Further, Fig. 1f is intriguing, but the authors do not comment much on it. I assume that the extreme colors indicate that there is no coexistence for these parameters, and thus that the region of coexistence is confined to the upper right hand area. Is it also obvious that the graph should be symmetric? Because the L and S strains are not interchangeable. Generally speaking, I think the authors pay too little mind to the coexistence ratio that is plotted in that figure, and in Fig. 4a. This ratio is predicted given the growth rate of each strain in their "native" environment, as well as the rate in the "opposite" environment, in a type of "payoff" matrix that is often used in Evolutionary Game Theory (EGT). Such a matrix can be seen, for example, in the article "Games microbes play" by Lenski and Velicer (2000). They are investigating the possible coexistence of two types "B" and "N" (never mind what these stand for), and write a matrix like the one I just discussed. So, for the strains S and L, you can have the fitness $w(L,L)$, which is the growth rate of L in an environment of only L, $w(S,S)$, which is the growth rate of S in an environment with only S (hence slower), as well as $w(L,S)$: the growth rate of L as a single bacterium on a lawn of S, and $w(S,L)$ vice versa.

When these four numbers are determined, the coexistence ratio (if there is coexistence) is uniquely determined, via the solution of the replicator equation. In the present case, this ratio would be $\frac{w(S,L)-w(L,L)}{w(S,L)-w(L,L)+w(L,S)-w(L,L)}$. If those rates had been easily accessible in the manuscript, I would have predicted coexistence ratios right here. Coexistence ratios as a function of biological parameters have been calculated in evolutionary game theory simulations for example in Adami et al., Phys. Rev. E 85 (2012) 011914.

Interestingly, in the EGT approach, it does not matter what kind of tradeoffs are giving rise to the negative frequency-dependent selection that enables the equilibrium. In fact, I was wondering why the authors do not use the words "negative frequency-dependent selection". The number is the metric should also determine the points where coexistence is not possible anymore.

In conclusion, I think this is an interesting study that demonstrates how it is possible to test hypotheses about evolutionary dynamics using computational simulations informed by parameters obtained experimentally. I think the authors can go quite a bit further in their analysis and predict coexistence ratios.

Reviewer #2 (Remarks to the Author):

LTEE experiments such as the one performed by Lenski are a great way to understand the evolutionary constraints that the organisms work with and the mechanisms behind the adaptation

to new environment. A key observation in these experiments is the spontaneous emergence of coexisting ecotypes that persist for a long time. One such example is the emergence of the L and the S strain in Lenski's experiment. Multiple studies have shown that the co-existence is due to trade-offs where one strain produces an overflow metabolite but can't use it and the other strain is good at using it.

This study from the Basan group have attempted to delineate the trade-offs behind the coexistence of ecotypes in LTEE. Specifically, they hypothesize and test if the growth rate-acetate excretion and the growth rate-lag time trade-offs are key to the ecotype co-existence.

Mathematical simulations and experimental studies were used nicely in a complementary way to test this hypothesis. The use of fluorescent strains to capture the growth dynamics during the diauxic shift (and hence the lag-phase dynamics) is a well thought out experiment.

The manuscript could benefit a lot if the following issues can be addressed:

1. I see that the key trade-off mechanistically is the one between glucose utilization and acetate utilization. A strain that uses glucose maximally cannot use acetate with equal efficiency. The trade-offs used in the study; lag-time, growth rate, and acetate excretion is downstream of the above-mentioned trade-off. The authors could comment on the advantages of using these trade-offs.
2. How does the growth rate-lag time-acetate excretion 3D plot look like? Since the authors already have the data, it will be useful to plot this and provide a single trade-off plot rather than 2 different ones. Moreover, the phrase "competing trade-offs" can be confusing. Why are the 2 trade-offs competing? If the authors prefer to use the 2 trade-offs model instead of a single one (3D) they should mention explicitly about both the trade-offs in the abstract. Currently it is missing the abstract.
3. While previous study has found that the growth rate between the L and the S strain vary by 20 %, why is it that the growth rates calculated here is not that different? Were there any changes in media between the two studies.
4. In the figure 1f the authors could show the frequency for lower growth rates where the co-existence breaks. How different should the growth rate be where the strains can no longer co-exist? This needs to be addressed in the text as well.
5. In Figure S3, the color coding for the L and S ecotype should be mentioned as the legend.
6. Line 166-168 needs to be reworded as the authors have not performed longitudinal measurements (time course) of phenotype (the growth rate and acetate excretion) for the two strains. The figure 3a-c rather shows a single snapshot of the phenotype and hence they cannot say that the phenotype is constrained for multiple generations.
7. In the frequency measurement experiments (fig 4) why is there a drop in frequency of the S strain before it starts to increase. This is at odds with simulations (figure 1e) where there is a monotonic change in the frequency.
8. The biochemical constraints and the phenotype of the L and S-strain varies across multiple generation in the LTEE as has been shown before by Mickaël Le Gac et al, (10.1073/pnas.1207091109). Given this, it is preferable to profile the growth rate and acetate excretion of the L and S strain from an earlier and later time-point (for example: 600 generations, 14000 generation and 40000 generations) and show that they do fall in the pareto-front. This will make the conclusions from the study more robust and generalizable.
9. The following references could be cited in appropriate places:
<https://doi.org/10.1073/pnas.1207091109> : This study has extensively profiled the coexistence dynamics of the L and S strain from different generations of Lenski's LTEE
<https://doi.org/10.1111/j.1558-5646.2008.00397.x>: This work shows that there is an order of events in the adaptation process. The lag phase adaptation precedes the growth rate adaptation.

- What are the noteworthy results?

While we know from the literature that the co-existence of the strains is due to trade-off of some kind and the glucose-acetate utilization dynamics is involved in it, the authors have specifically tested this hypothesis using both simulations and experiments. Capturing the lag-phase dynamics of the strains using fluorescently labelled strains is a noteworthy experiment. Combining the trade-off of growth rate-lag phase and growth rate- acetate excretion is also insightful.

- Will the work be of significance to the field and related fields? How does it compare to the established literature? If the work is not original, please provide relevant references.

The work will be of significance for analyzing coexistence dynamics of strains involving similar constraints in any LTEE experiments.

- Does the work support the conclusions and claims, or is additional evidence needed?

While the work does support the claims and the conclusions, the study can benefit a lot with additional experiments done on the L- and the S-strains from earlier and later generations. The conclusions then can be made more robust and generalizable. Similarly, additional plot showing the pareto surface combining all the three parameters would also help.

- Are there any flaws in the data analysis, interpretation and conclusions? Do these prohibit publication or require revision?

The data analysis is done well. The interpretation and conclusions can be more broader with additional experiments.

- Is the methodology sound? Does the work meet the expected standards in your field?

Both methodology and simulations used in the study is of high quality and meets the expected standards

- Is there enough detail provided in the methods for the work to be reproduced

Yes

Reviewer #3 (Remarks to the Author):

In the manuscript "Coexisting ecotypes in long-term evolution emerged from competing tradeoffs", Mukherjee et al.

addresses the observed coexistence of different strains in a long-term evolution experiment (LTEE) carried out in Lenski's

Lab over the last three decades. The experiment has elucidated many critical issues in evolutionary biology and

a rich phenomenology has emerged, urging us to provide new theories and understanding of the processes underneath.

The coexisting strains, referred to as L and S, were characterized phenotypically: The L strain displays an advantage over

the S strain in a medium provided with glucose. While using fermentation as the metabolic process, the L strain excretes

acetate. After sugar depletion, the strains must switch their metabolic pathways to growth on acetate. The S strain

is advantageous over the L strain by exhibiting a shorter lag time when

shifting from glycolytic to gluconeogenic (acetate) substrates. The S strain also presents a fitness advantage over

the L strain in the acetate medium.

In the paper, the authors claim that the coexistence between the two strains owes to the existence of two tradeoffs: the first, a tradeoff between

growth rate and acetate excretion, and the second, between lag time and growth rate. The problem is addressed

experimentally setup but also numerically. The paper is interesting but does not bring enough novelty

to deserve publication in Nature Communications. The manuscript is more appropriate to other journals such as Genetics, Evolution, Plos Computational Biology, and Journal of Evolutionary Biology.

Main point:

- The coexistence of two strains is not an outstanding finding in the experimental setup used.

The strains experience exponential growth during the fermentation phase, nothing happens during the time lag, and then there

is a period of slower growth while the acetate is consumed. This process is repeated as a fraction of the population after depletion of the acetate is used to inoculate tubes with fresh medium. Of course, resources mediate competition, with strain L

being a better competitor in the fermentation phase. In contrast, strain S becomes a better competitor in the medium with acetate. In resume,

one observes the coexistence of two species that compete for two limited resources. The outcome is in complete agreement with the competitive exclusion principle. The only novelty I see is that these two resources alternate over time. For instance, the experiment's outcome may change by increasing the duration of the fermentation phase.

Minor point:

- Can we refer to the relation between growth rate and acetate excretion (Eq. 1) as a tradeoff?

RESPONSE TO REVIEWERS' COMMENTS

Reviewer #1

*In this contribution, the authors use a computational simulation of bacterial growth to demonstrate that the coexistence of two evolved strains of *E. coli* bacteria (obtained from the LTEE) can coexist due to competing tradeoffs. I find the study to be well done, the arguments are convincing, and the computational simulation appears to be working well.*

We thank the reviewer for carefully reading our manuscript and many helpful suggestions that we think have improved our manuscript substantially. We address the individual questions in our point-by-point below.

There are a number of things that I would like the authors to clarify. And I also have a suggestion for a type of analysis that might clarify some aspects of the dynamics.

First, I was wondering why the authors performed the evolutionary simulation only in triplicate. I would imagine that with today's computers you could get much larger numbers, so that it is possible to quantitatively test a hypothesis, as opposed to writing: "it converged each time".

Yes, this is an excellent point. Especially because the simulation can be dominated by slow dynamics of incrementally different strains. It is also important to scan for any dependence on the initial configuration of strains.

To address these issues, we have now run the evolutionary simulation 10,000 times, each time initialized with random initial growth rates to scan the space of possible initial configurations. Indeed, these simulations converge to the same combination of strains. These data suggest that this combination of strains is the sole stable fixed-point of the evolutionary dynamics.

These new results are presented in Fig. 2d,e and Fig. S2b,c,d.

Further, Fig. 1f is intriguing, but the authors do not comment much on it. I assume that the extreme colors indicate that there is no coexistence for these parameters, and thus that the region of coexistence is confined to the upper right hand area. Is it also obvious that the graph should be symmetric? Because the L and S strains are not interchangeable. Generally speaking, I think the authors pay too little mind to the coexistence ration that is plotted in that figure, and in Fig. 4a.

We apologize. This plot should have been discussed in more detail. We have added additional explanation in our revised manuscript and think the discussion of this plot is now clearer.

The color code represents the frequency of strain #2. Therefore, a frequency of either 1 or 0 means there is no coexistence because there is only one strain remaining. This is the case everywhere for low growth rates. This is because there is no acetate excretion at slow growth (Eq. 1). Therefore, in this regime, the faster growing strain of the two takes over the population and there is no coexistence.

The plot is symmetric, because the two strains are interchangeable as we assume that their phenotypes are fully defined by their growth rates according to Eqs. [1-2]. The growth rates of the two strains are plotted on the axes and therefore the plot must be symmetric.

These growth rates determine acetate excretion rate and lag time according to the two trade-offs given by Eqs. [1-2].

With this approach, we do not *a priori* define which strain is the L strain and the S strain. Instead, in certain parameter regimes there is coexistence of a faster growing and a slower growing strain that can then be thought of as the L strain and the S strain.

We have tried to clarify these important points by adding additional details in our revised manuscript.

This ratio is predicted given the growth rate of each strain in their "native" environment, as well as the rate in the "opposite" environment, in a type of "payoff" matrix that is often used in Evolutionary Game Theory (EGT). Such a matrix can be seen, for example, in the article "Games microbes play" by Lenski and Velicer (2000). They are investigating the possible coexistence of two types "B" and "N" (never mind what these stand for), and write a matrix like the one I just discussed. So, for the strains S and L, you can have the fitness $w(L,L)$, which is the growth rate of L in an environment of only L, $w(S,S)$, which is the growth rate of S in an environment with only S (hence slower), as well as $w(L,S)$: the growth rate of L as a single bacterium on a lawn of S, and $w(S,L)$ vice versa.

When these four numbers are determined, the coexistence ratio (if there is coexistence) is uniquely determined, via the solution of the replicator equation. In the present case, this ratio would be $w(S,L)-w(L,L)/(w(S,L)-w(L,L)+w(L,S)-w(L,L))$. If those rates had been easily accessible in the manuscript, I would have predicted coexistence ratios right here. Coexistence ratios as a function of biological parameters have been calculated in evolutionary game theory simulations for example in Adami et al., Phys. Rev. E 85 (2012) 011914.

We appreciate the comments from the reviewer pointing out this interesting literature. We have cited these works in our revised manuscript and tried to do a better job discussing our work in the context of these previous findings.

However, we would like to point out that in our mechanistic model, exact solutions for the coexistence frequencies are not easy to derive analytically. This is mainly because the model consists of a set of transcendental equations for which no analytical solutions can be written. Specifically, some unknowns are in the exponents like the timescale of carbon depletion and some unknowns are pre-factors like the abundances of the strains.

We decided to focus on the simulation for this reason.

The approach mentioned by the reviewer cannot easily be applied here, because the fitnesses $w(L,S)$ is not constant, but a complicated function of the abundances of both L and S. For example, how much acetate is produced depends on how much of each strain is present. It is possible to determine the fitness values from the simulation after it has converged and apply the expression $w(S,L)-w(L,L)/(w(S,L)-w(L,L)+w(L,S)-w(L,L))$ without having to solve the complete dynamics. But crucially this is only possible *a posteriori* after one knows the results of the simulation and the fixed point of the dynamics. This is at least our understanding, but we are not experts in EGT.

Interestingly, in the EGT approach, it does not matter what kind of tradeoffs are giving rise to the negative frequency-dependent selection that enables the equilibrium. In fact, I was wondering why the authors do not use the words "negative frequency-dependent selection". The number is the metric should also determine the points where coexistence is not possible anymore.

In our revised manuscript, we have tried to better put our specific model in context with this more general body of work, including the concept of negative frequency-dependent selection.

In conclusion, I think this is an interesting study that demonstrates how it is possible to test hypotheses about evolutionary dynamics using computational simulations informed by parameters obtained experimentally. I think the authors can go quite a bit further in their analysis and predict coexistence ratios.

As mentioned above, to our understanding it is not possible to derive an exact analytical solution for coexistence ratios.

Reviewer #2

LTEE experiments such as the one performed by Lenski are a great way to understand the evolutionary constraints that the organisms work with and the mechanisms behind the adaptation to new environment. A key observation in these experiments is the spontaneous emergence of coexisting ecotypes that persist for a long time. One such example is the emergence of the L and the S strain in Lenski's experiment. Multiple studies have shown that the coexistence is due to trade-offs where one strain produces an overflow metabolite but can't use it and the other strain is good at using it.

This study from the Basan group have attempted to delineate the trade-offs behind the coexistence of ecotypes in LTEE. Specifically, they hypothesize and test if the growth rate-acetate excretion and the growth rate-lag time trade-offs are key to the ecotype co-existence.

Mathematical simulations and experimental studies were used nicely in a complementary way to test this hypothesis. The use of fluorescent strains to capture the growth dynamics during the diauxic shift (and hence the lag-phase dynamics) is a well thought out experiment.

We thank the reviewer for carefully reading our manuscript and making many insightful comments that have helped us improve our manuscript. We have marked the changes in our revised manuscript. Below is a point-by-point response to the reviewers' suggestions.

1. I see that the key trade-off mechanistically is the one between glucose utilization and acetate utilization. A strain that uses glucose maximally cannot use acetate with equal efficiency. The trade-offs used in the study; lag-time, growth rate, and acetate excretion is downstream of the above-mentioned trade-off. The authors could comment on the advantages of using these trade-offs.

This is an important point and we have tried to clarify this in our revised manuscript.

Utilization of acetate is not simultaneous with glucose utilization, but in sequence, after glucose has been depleted. We have shown this to be true also experimentally for the L and S strain, because both the L strain and the S strain excrete acetate while glucose is present (see Fig. 3b). There is a time delay before acetate can be used called the lag time that depends on the growth rate on glucose, as has been demonstrated in Ref. 13. The faster the growth, the longer the delay. This is one of the two trade-offs, but it is not downstream of the other trade-off.

The other mandatory trade-off is that the faster growth results in more acetate excretion during growth on glucose. This trade-off is given by Eq. [1] that has been established in Ref. 11.

Both trade-offs are independent of each other and both trade-offs are required for coexistence. When no acetate is excreted (e.g. below the acetate excretion threshold growth rate), there is no coexistence (see left, lower side of Fig. 1f for simulation and triangles in Fig. 4b, c for experimental data). In other words, if the acetate excretion trade-off did not exist, a fast-growing strain could evolve that does not excrete acetate and completely utilizes glucose. This strain would take over the population, because it would grow fast and would have no need to switch to acetate.

Similarly, if the two strains had the same lag time and growth rate on acetate, the faster growing strain on glucose would also take over the population and again there would be no coexistence.

Hence, both of these trade-offs are required for evolutionarily stable coexistence.

2. How does the growth rate-lag time-acetate excretion 3D plot look like? Since the authors already have the data, it will be useful to plot this and provide a single trade-off plot rather than 2 different ones. Moreover, the phrase "competing trade-offs" can be confusing. Why are the 2 trade-offs competing? If the authors prefer to use the 2 trade-offs model instead of a single one (3D) they should mention explicitly about both the trade-offs in the abstract. Currently it is missing the abstract.

We have added a 3D plot of the region accessible according to the two trade-offs (see Fig. S1). Each Pareto front is given by a 2D surface in the 3D phenotypic space. Hence, the intersection of these two Pareto fronts is a line plotted in Fig. S1.

This is a good suggestion. To avoid confusion, we have changed the formulation from “competing trade-offs” to “interacting trade-offs”. We think this is clearer, because the two trade-offs are interacting because acetate excreted according to the first trade-off leads to a diauxic shift when with a lag time given by the second trade-off.

As suggested, we have also tried to clarify the explanation of the two trade-offs in the abstract.

3. While previous study has found that the growth rate between the L and the S strain vary by 20 %, why is it that the growth rates calculated here is not that different? Were there any changes in media between the two studies.

Yes, there are subtle differences in the growth medium. To narrow down the mechanistic origin of coexistence, we wanted to rule out the role of other components of the medium in the LTEE. Therefore, we decided to use a simpler medium formulation, N⁺C⁺ minimal medium with glucose as the only carbon source. Specifically, there is no citrate in our medium. As demonstrated in Fig. 4, this does not affect the emergence of coexistence of the L and S strain. Despite those small difference in the medium, we are skeptical that these will result in different relative growth rates.

In our hands, the difference in growth rates was 13% as measured in four biological replicates in batch culture by taking 4-6 samples and measuring OD600 along the growth curve. We consider this to be the gold standard method of determining growth rate.

Even with this method, a difference between 13% and 20% growth rate difference is difficult to resolve experimentally. We are also not totally certain about the exact protocol how growth rates were determined in the original study.

4. In the figure If the authors could show the frequency for lower growth rates where the coexistence breaks. How different should the growth rate be where the strains can no longer coexist? This needs to be addressed in the text as well.

We have added additional clarifying remarks in the main text. One simple answer is that when both strains grow at slow growth rates below the acetate excretion threshold given by λ_0 , there is no acetate excretion according to Eq. [1] of the main text.

No general answer can be given when one of the strains produces acetate. Coexistence may emerge or the faster growing strain may still outcompete the slower growing one, as can be seen by the large regime in Fig. 1f in which there is no coexistence. Whether or not there is coexistence depends on the specific combination of parameters.

5. In Figure S3, the color coding for the L and S ecotype should be mentioned as the legend.

We have added this information to the caption and legend of Fig. S3.

6. Line 166-168 needs to be reworded as the authors have not performed longitudinal measurements (time course) of phenotype (the growth rate and acetate excretion) for the two strains. The figure 3a-c rather shows a single snapshot of the phenotype and hence they cannot say that the phenotype is constrained for multiple generations.

This is a good point. We have changed the discussion of Fig. 3 in the main text. We hope that this is now more clearly reflected in our revised manuscript.

7. In the frequency measurement experiments (fig 4) why is there a drop in frequency of the S strain before it starts to increase. This is at odds with simulations (figure 1e) where there is a monotonic change in the frequency.

Frequency only gives the abundance of the two strains relative to each other. However, the total abundances can also change as the experimental system finds the stable configuration and these dynamics are more complicated. Indeed, in the beginning of our coexistence experiment, we observe that total colony number changes over the first days of the experiment. This indicates that our initial inoculum abundance was not ideal.

Note that in the original paper by the Rozen and Lenski, they saw a monotonic change in frequency.

8. The biochemical constraints and the phenotype of the L and S-strain varies across multiple generation in the LTEE as has been shown before by Mickaël Le Gac et al, (10.1073/pnas.1207091109). Given this, it is preferable to profile the growth rate and acetate excretion of the L and S strain from an earlier and later time-point (for example: 600 generations, 14000 generation and 40000 generations) and show that they do fall in the Pareto-front. This will make the conclusions from the study more robust and generalizable.

This is indeed a very interesting point. On long timescales, the coexistence of the L and S strain in the LTEE exhibits additional complex dynamics. There are many possible reasons, but one explanation within the context of our model is that because the two trade-offs originate from resource allocation, the loss of large protein sectors that are not required for growth (e.g. flagella), while not abolishing the trade-offs, can lead to faster growth rates and shift the entire Pareto fronts. This has been demonstrated experimentally, e.g. Ref. 9. Such effects may lead to complex evolutionary dynamics. However, trade-off relations would need to be mapped out for each strain. We plan to investigate these fascinating long-term evolutionary dynamics and characterize how the Pareto fronts are shifted over the course of evolution in a follow-up work, but we think this is beyond the scope of the current study.

9. The following references could be cited in appropriate places:

<https://doi.org/10.1073/pnas.1207091109> : This study has extensively profiled the coexistence dynamics of the L and S strain from different generations of Lenski's LTEE

<https://doi.org/10.1111/j.1558-5646.2008.00397.x>: This work shows that there is an order of events in the adaptation process. The lag phase adaptation precedes the growth rate adaptation.

Thank you for pointing out these references that help putting our work in context. We have added these references and a short discussion of each in our revised manuscript.

Reviewer #3

In the manuscript "Coexisting ecotypes in long-term evolution emerged from competing tradeoffs", Mukherjee et al. addresses the observed coexistence of different strains in a long-term evolution experiment (LTEE) carried out in Lenski's Lab over the last three decades. The experiment has elucidated many critical issues in evolutionary biology and a rich phenomenology has emerged, urging us to provide new theories and understanding of the processes underneath.

The coexisting strains, referred to as L and S, were characterized phenotypically: The L strain displays an advantage over the S strain in a medium provided with glucose. While using fermentation as the metabolic process, the L strain excretes acetate. After sugar depletion, the strains must switch their metabolic pathways to growth on acetate. The S strain is advantageous over the L strain by exhibiting a shorter lag time when shifting from glycolytic to gluconeogenic (acetate) substrates. The S strain also presents a fitness advantage over the L strain in the acetate medium.

In the paper, the authors claim that the coexistence between the two strains owes to the existence of two tradeoffs: the first, a tradeoff between growth rate and acetate excretion, and the second, between lag time and growth rate. The problem is addressed experimentally setup but also numerically. The paper is interesting but does not bring enough novelty to deserve publication in Nature Communications. The manuscript is more appropriate to other journals such as Genetics, Evolution, Plos Computational Biology, and Journal of Evolutionary Biology.

Main point:

- The coexistence of two strains is not an outstanding finding in the experimental setup used. The strains experience exponential growth during the fermentation phase, nothing happens during the time lag, and then there is a period of slower growth while the acetate is consumed. This process is repeated as a fraction of the population after depletion of the acetate is used to inoculate tubes with fresh medium. Of course, resources mediate competition, with strain L being a better competitor in the fermentation phase. In contrast, strain S becomes a better competitor in the medium with acetate. In resume, one observes the coexistence of two species that compete for two limited resources. The outcome is in complete agreement with the

competitive exclusion principle. The only novelty I see is that these two resources alternate over time. For instance, the experiment's outcome may change by increasing the duration of the fermentation phase.

We think that there has been a misunderstanding here, which has resulted in the big discrepancy with the assessment of the other two reviewers. Based on these comments, we have tried to revise our manuscript to make these points clearer. We hope the comments below provide some clarification.

Note there is only one carbon resource provided in the medium, not two as the reviewer stated above. Glucose is the only carbon provided. Acetate is only produced by the bacterium during growth on glucose and not provided in the medium. Therefore, there are two strains stably coexisting on a single resource. Although this is not a major point of our work, this situation does violate the competitive exclusion principle.

Since acetate is produced by the bacteria themselves during growth on glucose, why doesn't a fast-growing strain evolve that simply does not produce acetate? Such a strain would easily take over the population and abolish coexistence. The first trade-off explains why such a strain cannot emerge.

Why doesn't a fast-growing strain emerge that can immediately grow on acetate? Such a strain would also easily outcompete the other strain and take over the population. The second trade-off explains why this strain doesn't appear.

The reviewer has not provided an alternative explanation for the emergence and stability of coexistence of the L and S strain. If the reviewer has such an explanation consistent with the data, it would be great if the reviewer would explain.

Similarly, if the reviewer is aware of previous work that has already explained the evolutionarily stable coexistence of the L and S strain (which we are unaware of), it would be great if the reviewer could kindly point out specific references.

Minor point:

- Can we refer to the relation between growth rate and acetate excretion (Eq. 1) as a tradeoff?

Yes, Eq. 1 is a precisely a trade-off. Faster growth is only possible at the cost of acetate excretion. Acetate excretion is a cost because it results in incomplete utilization of the carbon source and a lower biomass yield at the timepoint of glucose depletion. Without this relationship, there would be no reason for acetate to be excreted in the LTEE, as bacteria could simply evolve with fast growth rates but no acetate excretion. In this case there would be no coexistence.

REVIEWERS' COMMENTS

Reviewer #1 (Remarks to the Author):

I am satisfied with the author's response and changes to the manuscript.

Reviewer #2 (Remarks to the Author):

The authors have systematically addressed all the comments/queries. I recommend that the manuscript be accepted as is.

Reviewer #3 (Remarks to the Author):

It is pretty clear from my previous report that I referred to the acetate as a metabolic byproduct of Sugar, which in their case, is produced by the L strain. So, I kindly ask the Authors not to change the meaning of my words in the report (please see the second paragraph of my previous report below):

"The coexisting strains, referred to as L and S, were characterized phenotypically: The L strain displays an advantage over the S strain in a medium provided with glucose. While using fermentation as the metabolic process, the L strain excretes acetate. After sugar depletion, the strains must switch their metabolic pathways to growth on acetate. The S strain is advantageous over the L strain by exhibiting a shorter lag time when shifting from glycolytic to gluconeogenic (acetate) substrates. The S strain also presents a fitness advantage over the L strain in the acetate medium."

Regarding the references discussing the coexistence of two strains in a scenario proposed by the Authors, I mention the work of R. C. MacLean and I. Gudelj, *Nature* 441, 498 (2006). From a theoretical perspective, I suggest Fernández et al—Coexistence of competing metabolic pathways in well-mixed populations, *PHYSICAL REVIEW E* 93, 052401 (2016). There are many other ways in which a coexistence between two strains can be found. For a recent study, please see Wortel—Evolutionary coexistence in a fluctuating environment by specialization on resource level (<https://doi.org/10.1101/2021.05.18.444718>) and reference therein.

My recommendation for rejecting the manuscript for publication in *Nature Communications* relies on my perception that the current contribution does not match the criteria of novelty and impact. In my opinion, the paper seems more appropriate for publication in *Scientific Reports*.